# Solid Organ Transplantation Is Associated with an Increased Rate of Mismatch Repair Deficiency and PIK3CA Mutations in Colorectal Cancer

Eric S. Christenson [1], Valerie Lee [1], Hao Wang [1], Mark Yarchoan [1], Ana De Jesus-Acosta [1], Nilo Azad [1], Ahmet Gurakar [2], Ming-Tseh Lin [3], Dung T. Le [1], Daniel C. Brennan [4], Elizabeth M. Jaffee [1] and Katherine Bever [1,*]

1   Sidney Kimmel Comprehensive Cancer Center at Johns Hopkins, Department of Oncology, Johns Hopkins University, Baltimore, MD 21287, USA
2   Division of Gastroenterology, Department of Medicine, Johns Hopkins University, Baltimore, MD 21287, USA
3   Department of Pathology, Johns Hopkins University, Baltimore, MD 21287, USA
4   Department of Medicine, Johns Hopkins University, Baltimore, MD 21287, USA
*   Correspondence: kbever1@jhmi.edu; Tel.: +1-443-287-0966

**Abstract:** Solid organ transplants are associated with a modestly increased risk of colorectal cancers (CRC). However, the molecular profile of these cancers has not been described. We hypothesized that transplant-related immunosuppression may promote development of more immunogenic tumors as suggested by a high tumor mutation burden or mismatch repair deficiency. We performed an electronic medical record search for patients seen in the Johns Hopkins University Health System (JHHS) between 2017 and 2022 who developed CRC following solid organ transplantation. A comparator cohort of patients treated for CRC at JHHS with molecular profiling data was also identified. In this case, 29 patients were identified that developed post-transplant CRC (renal transplant, *n* = 18; liver transplant, *n* = 8; kidney-liver transplantation, *n* = 3). Compared to the JHHS general population CRC cohort, patients who developed post-transplant CRC had a higher rate of mismatch repair deficiency (41% versus 12%, *p*-value = 0.0038), and elevated tumor mutation burden (median of 22 mut/Mb versus 3.5 mut/Mb, *p*-value = 0.033) (range 3.52–53.65). Post-transplant tumors were enriched for PIK3CA mutations (43% versus 24%, *p*-value = 0.042). Post-Transplant CRCs are associated with clinical and molecular features of immune sensitivity, supporting a potential role for impaired immune surveillance in shaping the landscape of CRCs. These results may help inform the management of patients with post-transplant CRC.

**Keywords:** mismatch repair deficient colon cancer; PIK3CA mutations; calcineurin Inhibitors; post-transplant malignancies; high tumor mutation burden tumors; immune surveillance

## 1. Introduction

Solid organ transplantation represents an increasingly popular approach for the treatment of organ dysfunction [1,2]. These treatments are made possible through advanced immunosuppression strategies that allow the body to acclimate to the transplanted organ with manageable levels of rejection [3,4]. Unfortunately, this reduced immune surveillance comes at the cost of impaired anti-tumor immune activity, and many cancers are noted to occur at increased incidences in this post-transplant population [5–8]. Previous epidemiologic work has suggested that tumor types associated with the highest tumor mutation burdens (TMBs) tend to exhibit the largest increases in incidence following transplantation supporting a role of the immune system in this increased cancer rate [9].

While the incidence of colorectal cancer (CRC) is modestly increased in post-transplant registry data, it is unknown whether these excess cancers differ phenotypically from those in immunocompetent hosts [6,9]. We hypothesized that CRC arising in patients with a history

of transplant will have a higher mutational burden and distinct molecular profile resulting from reduced immune-mediated neoantigen loss compared to historical immunocompetent controls. In order to explore this question, we evaluated a retrospective cohort of patients treated within the Johns Hopkins Health System (JHHS) at Johns Hopkins Hospital or Sibley Memorial Hospital with a diagnosis of CRC and a known transplant history.

## 2. Materials and Methods

Records of patients noted to have a diagnosis of a liver, lung, heart, or kidney transplant in combination with colon adenocarcinoma or rectal adenocarcinoma from the years 2017 to 2022 were identified using the EPIC slicer dicer search function under a JHHS IRB-approved protocol (IRB00309275). Patient medical records were manually reviewed to determine the timing of transplant and CRC development. Any patients found to have CRC diagnosed prior to transplant were excluded. Patient information was then extracted including transplant type and year, stage of colorectal cancer, treatment course, molecular background of tumor, TMB, mismatch repair status, and outcomes data. Molecular sequencing results were recorded when available and used for analysis.

Comparisons were made between this post-transplant cohort and cohorts of general population CRC patients at Johns Hopkins Hospital that underwent TMB, mismatch repair, and/or mutational profiling. For these investigations, we used previously collected general population cohorts of patients with CRC that had undergone TMB and mismatch repair status testing ($n$ = 266) from January 2019 to December 2019 and molecular profiling on our in-house next-generation sequencing platform ($n$ = 544) from September 2017 to Decemeber 2019. Right-sided versus left-sided tumor prevalence was compared to the Swedish Family-Cancer Database which was a population of 6105 patients broken down by tumor location [10]. Proportions of patients that are mismatch repair proficient/deficient (pMMR/MMRd), right versus left-sided tumors, and with specific molecular alterations were compared to the general population using Fisher's exact testing.

## 3. Results

A total of 29 patients with solid organ transplantations and subsequent CRC diagnosis were identified. In this case, 18, 8, and 3 patients had prior renal, liver, and kidney and liver transplant, respectively. No cases of post-heart or lung transplant CRC were found in our cohort. The median age at colon cancer diagnosis was 60 years old (range: 31 to 82). In this case, 13 of these patients were female and 16 were male. Here, 18 of these patients were Caucasian, 9 were African-American, 1 was Pacific Islander, and 1 was Asian-American. These patients presented at a spectrum of cancer stages: stage 0 disease (1 patient), stage 1 disease (6 patients), stage 2 disease (9 patients), stage 3 disease (10), stage 4 (3). Of note, seven of the patients with initially earlier stage disease (one stage 1, four stage 2, two stage 3) subsequently recurred with stage 4 disease. The time from solid organ transplant to CRC diagnosis ranged from 2 to 35 years with a median time of 9 years. Of note, patients underwent pre-transplant colonoscopy per American Cancer Society screening guidelines. Only two of the patients in this cohort developed CRC within 10 years of transplant and under age 50 at 6 and 7 years post-transplant, respectively, suggestive that carcinogenesis of these lesions started prior to transplantation. The immunosuppression regimen at CRC diagnosis varied between patients with 24 of the 29 (83%) receiving calcineurin inhibitors. These demographic characteristics are outlined in Table 1.

In order to evaluate whether prior solid organ transplantation results in differences in CRC presentation or molecular background, we evaluated the microsatellite instability or mismatch repair status, tumor mutational burden, and mutation profile. A total of 17 (59%) patients had tumors that had undergone mismatch repair testing as part of their work up. Of these, 7 were MMRd while the other 10 were pMMR. Using a Fisher's exact test to compare the ratio of MMRd CRC (41%; 95% CI, 18–67%) versus the JHH non-transplant CRC cohort (12%), there was significant enrichment of MMRd tumors in the transplant population ($p$-value = 0.0038). TMB was also available for a total of 10 patients (5 MMRd,

5 pMMR) with median TMB of 22.0 (95% CI, 11.3–32.7) (range 3.52–53.65) within this population. pMMR tumors had median TMB of 14 (range 3.52–29) and MMRd tumors had a median TMB of 37 (range 14.1–53.7). Using a two-tailed students t-test to compare our transplant cohort to the general CRC population (median TMB of 3.5) showed significantly higher TMB in our post-transplant group (*p*-value = 0.033). In order to determine if the duration of immunosuppression influenced the frequency of MMRd tumors and tumor mutation burden, the time from transplant to CRC diagnosis was compared to MMRd status and TMB. There was no significant association between time from transplant to CRC diagnosis and mismatch repair status. There was a trend towards higher TMB in patients that developed CRC sooner after transplant but this did not reach significance (Supplemental Figure S1).

**Table 1.** The demographics of Post-Transplant Patients that Developed Colorectal Cancer.

| | | All Patients (*n* = 29) | Kidney (*n* = 18) | Liver (*n* = 8) | KLT (*n* = 3) |
|---|---|---|---|---|---|
| Age in years (range) | | 60 (31–82) | 51 (31–77) | 62.5 (54–73) | 73 (66–82) |
| Sex (%) | Male | 16 (55) | 10 (56) | 4 (50) | 2 (67) |
| | Female | 13 (45) | 8 (44) | 4 (50) | 1 (33) |
| Time from transplant to CRC diagnosis in years | | 9 (2–35) | 11 (3–35) | 3.5 (2–13) | 6 (2–12) |
| TMB median (range) | | 21.98 (3.52–53.7) | 14.07 (3.52–31.7) | 45.8 (37.8–53.7) | 37 |
| Mismatch Repair Status (%) | MMRd | 7 (41) | 3 (27) | 2 (50) | 2 (100) |
| | pMMR | 10 (59) | 8 (63) | 2 (50) | 0 |
| ALC at CRC dx (K/µL) median/range | | 1.1 (0.25–2.99) | 1.2 (0.48–2.99) | 0.84 (0.25–2.1) | 0.86 (0.83–1.43) |
| PIK3CA status (%) | Mutant | 6 (43) | 3 (38) | 2 (50) | 1 (50) |
| | Wild type | 8 (57) | 5 (62) | 2 (50) | 1 (50) |
| KRAS status (%) | Mutant | 3 (21) | 2 (25) | 0 | 1 (50) |
| | Wild type | 11 (79) | 6 (75) | 2 (100) | 1 (50) |
| BRAF status (%) | Mutant | 4 (31) | 1 (14) | 2 (50) | 1 (50) |
| | Wild type | 9 (69) | 6 (86) | 2 (50) | 1 (50) |
| APC status (%) | Mutant | 5 (42) | 4 (57) | 0 | 1 (100) |
| | Wild type | 7 (58) | 3 (43) | 4 (100) | 0 |
| TP53 status (%) | Mutant | 5 (42) | 3 (43) | 2 (50) | 0 |
| | Wild type | 7 (58) | 4 (57) | 2 (50) | 1 (100) |
| Immunosuppression at the time of CRC diagnosis (%) | none | 2 (7) | 2 (11) | 0 | 0 |
| | CNI | 6 (21) | 1 (6) | 5 (62) | 0 |
| | CNI + MMF | 1 (3) | 0 | 1 (12) | 0 |
| | CNI + Pred | 7 (24) | 6 (33) | 0 | 1 (33) |
| | CNI + MMF + Pred | 9 (30) | 7 (39) | 0 | 2 (67) |
| | CNI + Pred + 6-MP | 1 (3) | 0 | 1 (12) | 0 |
| | mTORi + MMF | 1 (3) | 0 | 1 (12) | 0 |
| | mTORi + Pred | 1 (3) | 1 (6) | 0 | 0 |
| | AZA + Pred | 1 (3) | 1 (6) | 0 | 0 |

**Table 1.** *Cont.*

|  |  | All Patients (*n* = 29) | Kidney (*n* = 18) | Liver (*n* = 8) | KLT (*n* = 3) |
|---|---|---|---|---|---|
| Cause of Liver Transplant (%) | HCV | 3 (27) | N/A | 2 (25) | 1 (33) |
|  | ETOH | 3 (27) | N/A | 2 (25) | 1 (33) |
|  | NASH | 3 (27) | N/A | 2 (25) | 1 (33) |
|  | Primary Sclerosing Cholangitis | 1 (9) | N/A | 1 (12) | 0 |
|  | Alpha-1-Antitrypsin | 1 (9) | N/A | 1 (12) | 0 |
| Cause of Renal Transplant (%) | atypical HUS | 1 (5) | 1 (6) | N/A | 0 |
|  | CNI toxicity | 1 (5) | 0 | N/A | 1(33) |
|  | diarrhea 2/2 short gut | 1 (5) | 0 | N/A | 1 (33) |
|  | DM | 4 (19) | 4 (22) | N/A | 0 |
|  | DM/HTN | 1 (5) | 1 (6) | N/A | 0 |
|  | DM/FSGS | 1 (5) | 1 (6) | N/A | 0 |
|  | FSGS | 2 (10) | 2 (11) | N/A | 0 |
|  | GPA | 1 (5) | 1 (6) | N/A | 0 |
|  | HTN | 4 (19) | 3 (17) | N/A | 1 (33) |
|  | IgA nephropathy | 1 (5) | 1 (6) | N/A | 0 |
|  | oligomeganephronia | 1 (5) | 1 (6) | N/A | 0 |
|  | PKD | 1 (5) | 1 (6) | N/A | 0 |
|  | SLE | 2 (10) | 2 (11) | N/A | 0 |
| Stage at diagnosis (%) | Stage 0/1 | 7 (24) | 5 (28) | 2 (25) | 0 |
|  | Stage 2 | 9 (31) | 6 (33) | 1 (12) | 2 (67) |
|  | Stage 3 | 10 (34) | 5 (28) | 4 (50) | 1 (33) |
|  | Stage 4 | 3 (10) | 2 (11) | 1 (12) | 0 |
| Recurrence (%) | Yes | 7 (24) | 6 (33) | 0 | 1 (33) |
|  | No | 22 (76) | 12 (67) | 8 (100) | 2 (67) |
| Tumor location | Right | 20 (69) | 12 (67) | 5 (62) | 3 (100) |
|  | Left | 5 (17) | 3 (17) | 2 (25) | 0 |
|  | Rectal | 4 (14) | 3 (17) | 1 (12) | 0 |

AZA: azathioprine, CNI: calcineurin inhibitor, DM: Diabetes, ETOH: alcoholic, FSGS: focal segmental glomeru-losclerosis, HCV: hepatitis C, GPA: granulomatosis with polyangitis, HTN: hypertension, HUS: hemolytic uremic syndrome, MMF: mycophenolate mofetil, MMRd: mismatch repair deficient, pMMR: mismatch repair proficient, mTORi: mTOR inhibitor, NASH: non-alcoholic steatohepatitis, Pred: Prednisone, PKD: polycystic kidney disease, SLE: systemic lupus erythematosus.

We assessed whether these MMRd tumors could have resulted from undiagnosed Lynch syndrome. Of the patients with MMRd disease, 2 had BRAF-associated somatic hypermethylation of MLH1, the other 5 did not have germline testing results available.

As right-sided tumors tend to be associated with higher rates of mutagenesis and immune infiltration in the general population and were previously reported to be more common in post-transplant CRC, we next evaluated whether there was an increase in right-sided tumors (defined as cancer proximal to the splenic flexure compared to disease at the splenic flexure or more distal) [10,11]. Within the general population CRC cohort approximately 63% of patients had left-sided (distal) disease while the other 37% had right-sided (proximal) disease. Of the 29 patients with transplant-associated CRC, 20 were noted to have right-sided disease while only 9 had left-sided disease. Comparison between

non-transplant and the post-transplant cohorts were calculated using Fisher's exact test and demonstrated a significantly higher rate of right-sided colon cancers in the post-transplant population (*p*-value = 0.0007) (Figure 1).

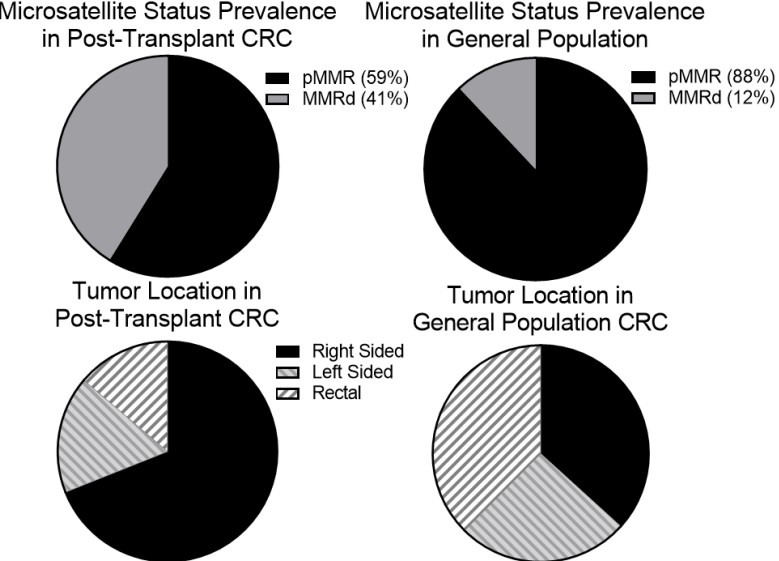

**Figure 1.** There is an increased proportion of mismatch repair deficient tumors and right-sided in the post-transplant population. We hypothesized that given the proposed role of the immune system in reducing relapse risk in mismatch repair deficient tumors many of these immunogenic tumors may be cleared before becoming clinically apparent. We suspected, therefore, that an increased proportion of tumors that developed in post-transplant patients would be mismatch repair deficient (MMRd). To test this, we identified 17 post-transplant CRC patients that with tumor mismatch repair testing. Of these, 7 (41%) were MMRd which is higher than the general population. Right-sided tumors also show higher degrees of tumor infiltrating lymphocytes suggesting that they may be better recognized by the immune system. We noted that our post-transplant population had a higher rate of right-sided tumors (20/29 or 69%) than the general population.

Previous renal transplant cohorts have reported prolonged CD4 lymphopenia in some patients following transplant-related immunosuppression [12–14]. We therefore looked at the absolute lymphocyte number (ALC) of patients at the time of their CRC diagnosis as a surrogate for CD4 lymphocyte count. We compared the ALC between pMMR (median: 1.29, mean: 1.55) and MMRd (median: 0.48, mean: 0.57) tumors using a student's T-test which showed a significantly lower ALC in patients who developed MMRd CRC (*p*-value = 0.009) (Figure 2). We also looked at TMB and ALC as continuous variables using Spearman's Rank-Order calculator. While there was a trend towards higher TMB in patients with lower ALC this did not meet statistical significance (*p*-value = 0.19). We next evaluated whether there were molecular differences between colorectal cancers from patients that underwent prior transplantation and those in the general population. In this case, 14 of the 29 patients had next-generation sequencing performed on their tumors. PIK3CA mutations were significantly enriched (Fisher's exact *p*-value = 0.042) in our transplant patients (6 of 14 patients), compared with 24% of CRCs profiled in the general population (Figure 3). In contrast, KRAS mutations trended toward being less common in our transplant cohort (3 out of 14) compared to 42% of the patients in the general population cohort (*p*-value = 0.17).

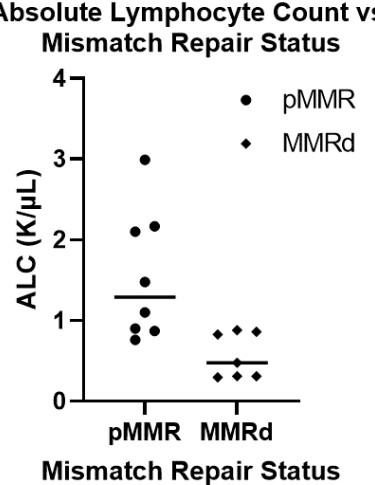

**Figure 2.** Comparing the absolute lymphocyte count between mismatch proficient and deficient colorectal cancer in transplant population. A total of 15 patients within our cohort had absolute lymphocyte counts (ALC) carried out at the time of their diagnosis along with mismatch repair status determination and tumor mutation burden. Of these, 8 patients were mismatch repair proficient (pMMR) and the other 7 patients were mismatch repair deficient (MMRd). The pMMR tumors had a higher ALC (median: 1.20, mean: 1.55) then the MMRd tumors (median: 0.48, mean: 0.57) which was statistically significant (student's t-test: *p*-value = 0.009).

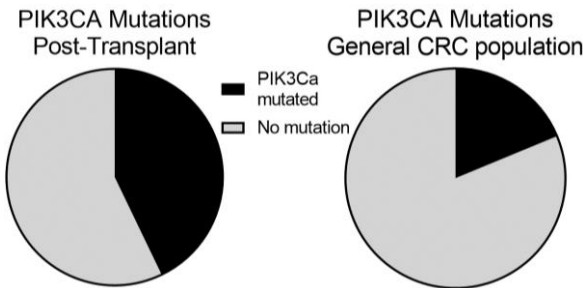

**Figure 3.** The distribution of PI3 Kinase-mTOR Mutations in Post-transplant and General Populations diagnosed with Colorectal Cancer (CRC). We identified 14 patients in our post-transplant population who had molecular profiling. Of them, 6 (44%; 95% CI, 18–71%) had PIK3CA which is significantly increased by Fisher's exact test (*p* = 0.042) compared to a 24% prevalence in a general CRC population.

## 4. Discussion

To our knowledge, this investigation of CRC in solid organ transplant recipients is the first to describe the molecular profile of some of these cancers and provides valuable insight into the potential roles that immunosuppression may play in CRC development. We report on 29 patients treated at JHHS who developed CRC after solid organ transplant. Similar to previous reports, we observed a higher proportion of proximal colon cancers in post-transplantation colon cancer compared to non-transplant CRC patients [15,16]. Right-sided disease has previously been shown to be associated with higher rates of tumor immune infiltration, MMRd disease (19.2%), and PIK3CA mutations (18.4%) all of which were noted to be enriched in our post-transplant CRC population [17,18].

Our data support the hypothesis that CRC that develops in transplant recipients is more immunogenic which may have implications for management. While the immune checkpoint inhibitor therapies have demonstrated excellent outcomes in certain immune-responsive cancers, including dMMR, their use in patients following solid tumor transplantation is not without significant risk of both graft loss and death [19–22]. One retrospective review from MD Anderson, identified 39 patients who received immune checkpoint therapy for treatment of cancer following solid organ transplant. Of these, 41%

experienced allograft rejection (11/23 renal, 4/11 hepatic, and 1/5 cardiac) while 15 of the 32 evaluable patients had tumor response to checkpoint therapy [19]. In a second report by Schenk et al., 8 patients with cutaneous malignancy following kidney transplant were treated with nivolumab +/− ipilimumab. Of these patients, 2 achieved a complete response to therapy. In this case, 3 experienced allograft loss despite being maintained on tacrolimus and prednisone during treatment including one that achieved a complete response [22]. An alternative strategy may be a reduction in immunosuppression in post-transplant patients with metastatic CRC. In particular, mycophenolate mofetil (MMF) is typically implicated in the development of lymphopenia, and its reduction or removal might represent a rational first step in patients with MMRd/high TMB tumors particularly those with concomitant lymphopenia.

The observation that the tumors that developed had increased PIK3CA mutations raises the possibility of immune-independent effects of immunosuppressants on CRC development. Specifically, calcineurin inhibitors can alter DNA damage repair in preclinical models through reduced expression of nucleotide excision repair and base excision repair pathways facilitating the acquisition of additional mutations that might promote carcinogenesis [23,24]. In addition to its possible role in excess mutagenesis, preclinical models suggest calcineurin inhibition with cyclosporine leads to increased TGF-β signaling in host cells resulting in a more invasive and mesenchymal phenotype [25]. Of note, this process is circumvented by mTOR inhibition which promotes increased apoptosis of malignant cells suggesting a role for PIK3CA-AKT in maintaining this process as an explanation for the excess mutations of this pathway in the post-transplant cohort. Whether mTOR inhibitors may be a more favorable option for transplant immunosuppression with lower risk of carcinogenesis is not known [26–28].

KRAS mutations were less common in our transplant population albeit not to the level of statistical significance. This was surprising because of the right-side predominance of CRC tumors in our post-transplant cohort and prior studies showing a higher frequency of KRAS mutations with CRC in this location [17,29]. One possible contributing factor is a slightly lower rate of KRAS mutations typically seen in MMRd disease. However, another potential explanation for this discrepancy is the established role of KRAS in facilitating tumor immune escape through downregulation of MHC-I [30]. In the absence of effective immune surveillance this function may be less critical leading to reduced selective pressure towards acquisition of KRAS mutations.

This observational cohort has several important limitations that must be considered when interpreting the outlined results. Foremost, this is a small retrospective study and therefore under-powered to draw many definitive conclusions. We used cohorts of general population patients with CRC who underwent molecular profiling at JHH as a comparator with the assumption that these population are comprised of immunocompetent individuals of a similar disease stage and environmental exposure profile. It is, however, possible that the factors that led to the need for solid organ transplantation or a change in lifestyle precipitated by receipt of an organ may have led to changes in the tumor mutation burden or mutation profile independent of the transplant itself. For example, obesity, history of tobacco or alcohol ingestion, and diabetes are all comorbidities that may have contributed both to a patient's need for a transplant and subsequent CRC development. Of note, the general population cohort of patients tested at JHH was chosen as this was felt to best control for other factors that might influence MMRd, TMB, or molecular profile testing both in terms of distribution of mutations such as ethnicity or stage of disease as well as institutional practice on who underwent these testing procedures. Reassuringly, our control population has a similar frequency of MMRd, PIK3CA mutations, KRAS mutations, and median TMB as was present in previously published cohort studies [31–33].

Another limitation was that due to the retrospective nature of our study, molecular testing results were not available on all patients, increasing the potential for bias in our results. In particular our patients that underwent this diagnostic testing tended to be those with recurrent and/or more advanced disease. Most of the patients that did not undergo

mismatch repair testing were early stage patients (1 intramucosal, 4 stage I, 1 stage II, and 6 stage III). This is consistent with MMRd testing having less impact on the clinical management of stage I and III disease. As the proportion of CRCs that are MMRd tends to be lower in patients with more advanced disease, the absence of this data for some patients likely does not account for the differences between our post-transplant patients and the general population.

## 5. Conclusions

Solid organ transplantation is increasingly employed to address a spectrum of medical conditions. Previous transplant databases showed a modest increase in CRC incidence amongst patients with kidney transplants. However, there is limited published data characterizing the tumors that emerge in this immunocompromised background [6,9]. Our data suggest that CRC arising in solid organ transplant patients is associated with an increased rate of PIK3CA mutations and MMRd disease. These features suggest that the increased risk of CRC may be due at least in part to immunosuppression used to prevent transplant rejection. While the use of immune checkpoint inhibitors in solid organ transplant recipients is generally contraindicated, reduction in immunosuppression may have a beneficial effect. Furthermore, our identification of higher rates of PIK3CA mutations may suggest that understanding the specific pathways that are altered with immunosuppressive agents may lead to new pathway specific immunosuppression that will reduce the increased risk for CRC and other cancers.

**Supplementary Materials:** The following supporting information can be downloaded at: https://www.mdpi.com/article/10.3390/curroncol30010006/s1, Figure S1: Association between Microsatellite Status, Tumor Mutation Burden and Time from Transplant to Colorectal Cancer diagnosis.

**Author Contributions:** Conceptualization, E.S.C., V.L., M.Y., A.D.J.-A., A.G., D.T.L., D.C.B., E.M.J. and K.B.; methodology, E.S.C., V.L. and K.B.; software, N.A.; validation, E.S.C., H.W. and K.B.; formal analysis, E.S.C., H.W. and K.B.; investigation, E.S.C., V.L. and K.B.; resources, M.-T.L.; data curation, E.S.C.; writing—original draft preparation, E.S.C., V.L. and K.B.; writing—review and editing, all authors; visualization, E.S.C.; supervision, D.T.L., D.C.B., E.M.J. and K.B.; project administration, E.S.C. All authors have read and agreed to the published version of the manuscript.

**Funding:** This research was generously supported by Swim Across America (ESC), Bloomberg-Kimmel Institute for Cancer Immunotherapy (All authors), Cancer Convergence Institute (All authors), Charles T. Bauer Foundation (DCB) and the Raymond and Melody Ranelli Fund (DCB). The APC was funded by Swim Across America.

**Institutional Review Board Statement:** The study was conducted according to the guidelines of the Declaration of Helsinki, and approved by the Institutional Review Board of Johns Hopkins University IRB00309275 12 June 2021.

**Informed Consent Statement:** Patient consent was waived by the Institutional Review Board due to the retrospective nature of this work and the challenges with consenting patients due to a high proportion having succumbed to their cancer/comorbidities or being otherwise unavailable for consent.

**Data Availability Statement:** The data that supports the findings of this study are available in Table 1 of this article. Additional information is available from the corresponding author upon request.

**Acknowledgments:** Thank you to Daniel Laheru, and Ben Park for your review of this manuscript and insightful feedback. This publication was made possible by the Johns Hopkins Institute for Clinical and Translational Research (ICTR) which is funded in part by Grant Number UL1 TR003098 from the National Center for Advancing Translational Sciences (NCATS) a component of the National Institutes of Health (NIH), and NIH Roadmap for Medical Research.

**Conflicts of Interest:** The authors declare no conflict of interest.

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
