# Peer review of "Solid Organ Transplantation Is Associated with an Increased Rate of Mismatch Repair Deficiency and PIK3CA Mutations in Colorectal Cancer"

_curroncol, doi:10.3390/curroncol30010006_

Round 1

Reviewer 1 Report

The article by Eric S. Christenson and others has described the some characteristics of colorectal cancer that occur as a secondary event to the use of immunosuppresion medications following solid organ transplant. It described these tumors by having a common mismatch repair deficiency and PIK3CA mutations. The article also described how these observations would help in deciding the therapeutic approach in those specific group of patients. The article is written with a good aim and effort, but the work has to improved.

- The major flaw in this work is sample size, it is quite small and cannot really draw conclusions and yield a strong significance. The authors agreed to that in their discussion and said that their data can be biased due to this main factor.

- The authors mentioned in their methods that they included patients with heart and lung transplants within their study cohort, however, I did not find any justifications about these patients and whether they developed CRC or not.

- The authors claimed that their work is providing a molecular profiling to CRC in solid organ transplant patients, but I do not find a real molecular profiling in the paper. They mentioned in the methods that they have analyzed NGS data from John Hopkins Hospital but yet no extensive NGS analysis of the data, or heatmaps to show key pathways affected. The authors mentioned almost only 2 main genetic changes, KRAS and PIK3CA, BRAF and other hereditary CRC gene status were not discussed to an adequate extent, but only one sentence about it.  

Author Response

Please see the attachment. We are grateful for your review.

Reviewer 2 Report

Dr Christenson et al detail a retrospective study on post-transplant colorectal cancer. It is novel and informative. I believe this report will be a valuable addition to the current literature. Just a few comments.

1.       Patients were diagnosed with CRC 2-35 years after solid organ transplant. Is it possible that those with CRC occurring early after transplant already had CRC at the time of transplant? If colonoscopy was routinely conducted before transplantation, this should be stated.

2.       Previous reports suggest increased risk of CRC in the early post-transplant period. A description and/or a figure/graph showing the distribution of time from transplant to CRC diagnosis may be interesting. Were CRCs diagnosed earlier after transplant associated more with MMRd or high TMB?

Author Response

Please see attachment. We are grateful for your review.

Reviewer 3 Report

Dear All,

I was pleased to review the article “Solid Organ Transplantation is Associated with an Increased Rate of Mismatch Repair Deficiency and PIK3CA mutations in Colorectal Cancer”.

The methodology used by the authors is appropriate for the purpose of the study and conclusions are narrowly linked to data discussion and available evidence.

This is a well-designed and a well-written article concerning the transplant-related immunosuppression and the possibility to promote development of more immunogenic tumors – CRC. In my opinion the content of the manuscript, its aim and the direction are clear. The manuscript is original and its topic is interesting.

The title expresses clearly the content of the manuscript and highlights the importance of the study. It doesn’t contain any unnecessary description.

The abstract is a short and clear summary of the aims, key methods, important findings and conclusions of the article and doesn’t contain unnecessary information. The introduction section clearly summarize the current state of the topic as well as clearly define the aim of the study. The introduction is this consistent with the rest of the manuscript.

Study design and methods are appropriate for the research question. The methods of selecting the appropriate publications has been described in detail.

The results are presented clearly and accurately and are consisted with the aim of the work and the methods. All the relevant data have been included in the article. The data described in the text are consistent with the data in the figures and tables.

The authors logically explain and describe their findings. The limitations of the study also have been described.

The authors cite the initial discoveries where suitable. The cited studies represent current knowledge.

I suggest to accept this paper.

Author Response

Please see the attachment. Thank you for taking the time to review our manuscript.

Round 2

Reviewer 1 Report

I still think that the article by Eric S. Christenson and others that discussed solid organ transplantation and their association with an increased rate of mismatch repair deficiency and PIK3CA mutations in colorectal cancer is still not suitable for publication.The authors should try to validate their findings by another study cohort and expand their research to find more molecular profiling details that would be applicable to the clinic and/or guide basic research in the field.

Reviewer 2 Report

The authors have adequately revised their manuscript. I have no further comments.